# Efficiency of Skeletal Muscle Mass/Weight Measurement for Distinguishing Metabolic Dysfunction-Associated Steatotic Liver Disease: A Prospective Analysis Using InBody Bioimpedance Devices

**DOI:** 10.3390/nu16244422

**Published:** 2024-12-23

**Authors:** Miwa Tatsuta, Tsutomu Masaki, Shungo Kimura, Yudai Sato, Akemi Tomida, Ichiro Ishikawa, Yu Nakamura, Kei Takuma, Mai Nakahara, Kyoko Oura, Tomoko Tadokoro, Koji Fujita, Joji Tani, Asahiro Morishita, Takashi Himoto, Shinjiro Miyazaki, Tsuyoshi Maeta, Yoshihiro Mori, Fumikazu Kohi, Hideki Kobara

**Affiliations:** 1Department of Gastroenterology, KKR Takamatsu Hospital, 4-18 Tenjinmae, Kagawa 760-0018, Japan; s-kimura@kkr-ta-hp.gr.jp (S.K.); y-sato@kkr-ta-hp.gr.jp (Y.S.); tomida@kkr-ta-hp.gr.jp (A.T.); 2Department of Gastroenterology and Neurology, Kagawa University School of Medicine, Kagawa 761-0793, Japan; takuma.kei@kagawa-u.ac.jp (K.T.); nakahara.mai@kagawa-u.ac.jp (M.N.); oura.kyoko@kagawa-u.ac.jp (K.O.); tadokoro.tomoko@kagawa-u.ac.jp (T.T.); fujita.koji@kagawa-u.ac.jp (K.F.); tani.joji@kagawa-u.ac.jp (J.T.); morishita.asahiro@kagawa-u.ac.jp (A.M.); kobara.hideki@kagawa-u.ac.jp (H.K.); 3Department of Gastroenterology, Kagawa Saiseikai Hospital, Kagawa 761-8076, Japan; tmasaki@saiseikai-kagawa.jp; 4Department of Neuropsychiatry, Kagawa University School of Medicine, Kagawa 761-0793, Japan; ishikawa.ichiro@kagawa-u.ac.jp (I.I.); nakamura.yu@kagawa-u.ac.jp (Y.N.); 5Department of Medical Technology, Kagawa Prefectural University of Health Sciences, Kagawa 761-0123, Japan; himoto@kagawa-puhs.ac.jp; 6Department of Rehabilitation, KKR Takamatsu Hospital, Kagawa 760-0018, Japan; miyazaki@kkr-ta-hp.gr.jp; 7Department of Internal Medicine, KKR Takamatsu Hospital, Kagawa 760-0018, Japan; maeta@kkr-ta-hp.gr.jp (T.M.); mori@kkr-ta-hp.gr.jp (Y.M.); kohi@kkr-ta-hp.gr.jp (F.K.)

**Keywords:** metabolic dysfunction-associated steatotic liver disease, bioimpedance analysis, body composition components, skeletal muscle mass, sarcopenia

## Abstract

Background/Objectives: Metabolic dysfunction-associated steatotic liver disease (MASLD) is diagnosed when hepatic steatosis is proven by imaging and one of the five cardiometabolic criteria is present. The relationship between MASLD and body composition components has recently received increased research attention. However, the five cardiometabolic criteria do not include components of body composition. This study aimed to identify significant body composition factors associated with MASLD in patients undergoing health checkups. Methods: This study included a cohort of 6599 examinees who participated in a health check-up conducted between 2022 and 2023, and their data were prospectively analyzed. The inclusion criteria were undergoing abdominal ultrasonography, alcohol consumption <30 g/day for males or <20 g/day for females, and one of the five cardiometabolic criteria. Results: Finally, 3864 examinees were enrolled. In total, 1133 (51.8%) males and 454 (27.1%) females had MASLD. Sarcopenia was present in only 0.62% of males and 0.66% of females with MASLD. The MASLD group had significantly lower skeletal muscle mass/weight (SMM/WT) values than the non-MASLD group. Multivariate analysis revealed that SMM/WT was independently associated with MASLD. Conclusions: SMM/WT was significantly associated with MASLD. Therefore, muscle mass assessment using SMM/WT may be a potential marker for diagnosing MASLD.

## 1. Introduction

Non-alcoholic fatty liver disease (NAFLD) is emerging as the most prevalent liver condition globally [1]. In 2023, it was proposed that conventional NAFLD and non-alcoholic steatohepatitis (NASH) should be diagnosed as metabolic dysfunction-associated steatotic liver disease (MASLD) and metabolic dysfunction-associated steatohepatitis (MASH), respectively, when at least one of the five cardiometabolic criteria is met. MASLD is diagnosed when hepatic steatosis is proven with imaging or biopsy, one of the cardiometabolic criteria is present, and no other factors, such as alcohol consumption, are present [2]. One of the main causes of MASLD is reduced physical activity levels. In 2007, obesity was reported as the primary contributing factor in the development of NAFLD/NASH. A positive correlation was shown between the amount of visceral and intrahepatic fat in patients with NAFLD/NASH [3]. On the other hand, Lee et al. demonstrated that sarcopenia is linked to an elevated risk of NAFLD, independent of obesity or metabolic control, in 2015 [4]. There is a close association between MASLD, lifestyle-related diseases, and sarcopenia. About 20–40% of patients with MASLD have sarcopenia [5,6]. MASLD and components of body composition, including muscle mass and fat mass, are closely and mutually involved in their respective pathologies. However, there are no reports on the body composition components that are most strongly associated with MASLD.

The Asian Working Group for Sarcopenia (AWGS) 2014 consensus defined sarcopenia as “the age-related decline in muscle mass, reduced muscle strength and/or diminished physical performance.” Additionally, they recommended using the skeletal muscle mass index (SMI; appendicular skeletal muscle mass divided by height squared), determined with bioelectrical impedance analysis (BIA), to assess muscle mass [7]. The AWGS 2019 guidelines also endorsed the use of SMI for assessing muscle mass through BIA [8]. In 2016, liver disease-specific secondary sarcopenia criteria were presented at the Japan Society of Hepatology (JSH). These criteria are used in many countries for a variety of diseases. The presence of sarcopenia is evaluated based on grip strength and muscle mass [9]. According to the JSH criteria (2nd edition), the threshold values for grip strength were set at 28 and 18 kg for males and females, respectively. The threshold values for muscle mass measurement by BIA were 7.0 and 5.7 kg/m^2^ for males and females, respectively [10].

Various skeletal muscle mass indices exist for assessing sarcopenia: appendicular skeletal muscle mass adjusted for height squared or weight [11]. In practice, SMI adjusted for height squared is often employed to assess muscle mass. On the other hand, skeletal muscle mass/weight (SMM/WT) is the total amount of skeletal muscle mass in the body, including the skeletal muscles of the trunk, divided by body weight and expressed as a ratio. It is an index used to evaluate skeletal muscle mass as a proportion of total body weight. One of the problems with the definition of sarcopenia is that the discordance between these indices and the indices for skeletal muscle mass have not yet been unified [11].

The five cardiometabolic criteria for MASLD are overweight, glucose intolerance, hypertension, hypertriglyceridemia, and hypo-high-density lipoprotein (HDL) cholesterolemia [2]. These cardiometabolic criteria do not include the components of body composition. Considering the components of body composition in MASLD is crucial, since many lean individuals with a body mass index (BMI) < 23.0 kg/m^2^ and low muscle mass have MASLD in Japan. The identification of body composition components may also positively impact planning an individualized MASLD exercise treatment program considering muscle mass and fat mass. In the present research, we examined the association between MASLD and body composition components, as well as the factors most strongly associated with MASLD among health checkup examinees at a single institution.

## 2. Materials and Methods

### 2.1. Study Population

This prospective observational study, conducted at a single center from April 2022 to March 2023, aimed to investigate the relationship between MASLD and body composition components, as well as identify the key factors most strongly associated with MASLD. We considered 6599 examinees who underwent health checkups at KKR Takamatsu Hospital. Examinees who fulfilled the following criteria were included: (1) undergoing abdominal ultrasonography, (2) alcohol consumption <30 g/day for males or <20 g/day for females, and (3) at least one cardiometabolic criterion. The cardiometabolic criteria were as follows: (1) BMI ≥ 23 kg/m^2^; (2) waist circumference > 94 cm for males or >80 cm for females; (3) fasting serum glucose ≥ 100 mg/dL; (4) glycated hemoglobin (HbA1c) of ≥5.7%; (5) treatment for type 2 diabetes; (6) systolic blood pressure of ≥130 mmHg; (7) diastolic blood pressure of ≥85 mmHg; (8) specific antihypertensive drug treatment; (9) plasma triglycerides ≥ 150 mg/dL; (10) plasma HDL cholesterol ≤ 40 mg/dL for males or ≤50 mg/dL for females; and (11) lipid-lowering treatment.

We enrolled 3864 examinees who met the inclusion criteria.

### 2.2. Diagnosis of Hepatic Steatosis by Abdominal Ultrasonography

We scored the findings of hepatic steatosis obtained from the B-mode abdominal ultrasound based on a report by Hamaguchi et al. [12]. A bright liver, a characteristic of hepatic steatosis-like conditions in which fat accumulation alters the acoustic properties of the liver, was defined on a scale from 0 to 2 points. Bright liver was diagnosed based on the presence of abnormally intense, high-level echoes originating from the hepatic parenchyma. The intensity was classified into three categories: none, mild, and severe. We defined a hepatorenal echo contrast, the relative brightness of the liver compared to the dark renal cortex (a significant contrast often signals liver abnormalities like steatosis), on a scale from 0 to 1 point. The diagnosis of hepatorenal echo contrast was determined by the apparent ultrasonographic contrast between the hepatic and right renal parenchyma, observed on a right intercostal sonogram at the mid-axillary line. Deep echo attenuation was defined on a scale from 0 to 2 points. The diagnosis of deep echo attenuation was determined by the noticeable reduction in echo penetration into the liver’s deeper regions and the impaired visualization of the diaphragm. Vessel blurring was defined on a scale from 0 to 2 points. Vessel blurring was diagnosed based on the inability to clearly visualize the borders of intrahepatic vessels and the narrowing of their lumen. Blurring of the boundary between the liver and the right kidney or gallbladder wall was defined on a scale from 0 to 2 points. Ultimately, we diagnosed hepatic steatosis with a total of ≥2 points of these scores [12]. Abdominal ultrasonography was conducted using two devices: the Xario 200 with a 6-MHz convex array transducer (Canon Medical Systems, Otawara, Japan) and the Aplio XG with a 6-MHz convex array transducer (Canon Medical Systems, Otawara, Japan).

### 2.3. Measurement of Body Composition Components, Hand Grip Strength, and Waist Circumference

We measured the body composition components of 6599 examinees who underwent health checkups. Body composition was evaluated using direct segmental multifrequency BIA (DSM-BIA) with the InBody770 device (InBody Co., Ltd., Seoul, Republic of Korea). The InBody770 utilizes a multi-frequency segmental measurement technique and employs an 8-point tactile electrode. Multi-frequency measurements were conducted using multiple frequencies at 1, 5, 50, 250, 500, and 1000 kHz for each body segment (arms, trunk, and legs). This analyzer demonstrated a strong correlation with the gold standard measurement, which utilizes dual-energy X-ray absorptiometry [13]. The DSM-BIA does not rely on statistical data from any particular population. As a result, body composition can be accurately assessed for individuals of diverse body types, including those with obesity, older adults, and others. The main research parameters using BIA are weight, BMI, total body water, intracellular water, extracellular water, proteins, minerals, body fat mass, soft lean mass, fat-free mass, skeletal muscle mass (SMM), percentage body fat, SMI, visceral fat area, SMM/WT, fat-free mass index, and fat mass index. Handgrip strength was measured in all 6599 examinees in the standing position using a Smedley hand dynamometer T.K.K.5401 (Takei, Niigata, Japan). Grip strength measurements were taken twice, alternating between the left and right sides, and higher values from each side were averaged. Waist circumference was measured at navel height in the relaxed standing position.

### 2.4. Statistical Analysis

The Wilcoxon signed-rank test was employed to analyze differences in characteristics between the non-MASLD and MASLD groups. The threshold for statistical significance was defined as *p* < 0.005. Multivariate analysis was conducted on the following eight key variables to identify factors associated with MASLD: BMI, waist circumference, total body water, proteins, percentage body fat, SMI, SMM/WT, and hand grip strength. We performed multiple linear regression analyses as sensitivity analyses. Logistic regression analysis was performed, with the threshold for statistical significance defined as *p* < 0.001. The analysis of these data was performed using JMP 16.2.0 (SAS Institute Inc, Cary, NC, USA).

## 3. Results

### 3.1. Prevalence of MASLD Among Health Checkup Examinees

A total of 3864 examinees (2188 males and 1676 females) were enrolled between April 2022 and March 2023. Abdominal ultrasonography revealed hepatic steatosis in 1133 males (51.8%) and 454 females (27.1%) (Figure 1).

### 3.2. Prevalence of Sarcopenia Among Health Checkup Examinees

Based on the JSH (2nd edition) criteria for sarcopenia, the prevalence of sarcopenia among examinees with MASLD was 0.62% and 0.66% in males and females, respectively (Figure 2).

### 3.3. Characteristics of the Non-MASLD Group and the MASLD Group

The characteristics of the non-MASLD and MASLD groups, along with their differences, are displayed in Table 1. Eighteen parameters (weight, BMI, waist circumference, total body water, intracellular water, extracellular water, proteins, minerals, body fat mass, soft lean mass, fat-free mass, SMM, percentage body fat, SMI, visceral fat area, fat-free mass index, fat mass index, and handgrip strength) were significantly higher in the MASLD group compared to the non-MASLD group in both males and females (*p* < 0.005). Contrastingly, only SMM/WT was significantly lower in both males and females in the MASLD group compared to the non-MASLD group (*p* < 0.0001). The SMM/WT distributions for the non-MASLD group and the MASLD group are shown in Appendix A.

### 3.4. Identification of Factors Strongly Associated with MASLD

We performed a multivariate analysis with eight major items: BMI, waist circumference, total body water, proteins, percentage body fat, SMI, SMM/WT, and hand grip strength. Waist circumference, percentage body fat, and SMM/WT showed significant differences in the multivariate analysis (*p* < 0.001). SMM/WT was the independent factor most strongly linked to MASLD, with an odds ratio of 5.11 (95% CI: 2.99–8.73) for males and 3.41 (95% CI: 1.68–6.93) for females (Table 2).

## 4. Discussion

This is the first report of a strong association between SMM/WT, among the body composition components, and MASLD. The AWGS and JSH recommend SMI for muscle mass assessment using BIA. SMM/WT is defined as skeletal muscle mass, including the skeletal muscles of the trunk, divided by weight. Contrastingly, SMI is defined as appendicular skeletal muscle mass divided by height squared. A correlation exists between muscle mass and body size, with larger individuals generally having more muscle mass [14]. Thus, skeletal muscle mass indices have been used to assess muscle mass, which is adjusted with various parameters, including height squared, weight, and BMI [11].

The appendicular skeletal muscle mass/height-squared ratio was suggested by Baumgartner et al. [15]. Since then, a significant number of studies have utilized this index to define sarcopenia. SMI is often used in actual practice to assess muscle mass. However, this indicator has some limitations. This index, adjusted for height squared, demonstrated a positive correlation with BMI. Consequently, individuals with higher fat mass and BMI are less likely to be diagnosed with sarcopenia [16].

The concept of the weight-adjusted muscle mass index was put forward by Janssen et al. According to this index, sarcopenia is strongly associated with functional impairment and disability [17]. This weight-adjusted muscle mass index is commonly used alongside the height-squared-adjusted index. Previous large-scale studies on NAFLD/NASH and sarcopenia have reported skeletal muscle mass adjusted for weight [18,19]. The weight-adjusted skeletal muscle mass index is an assessment of skeletal muscle mass as a percentage of body weight, which also considers body fat mass. Individuals with MASLD have larger fat areas and higher BMI. Therefore, SMM/WT is a more effective index than SMI for evaluating muscle mass in individuals with MASLD.

The meta-analysis in 2019 reported that the relative skeletal muscle mass in NAFLD patients was lower than that in healthy controls. However, the definition of relative skeletal muscle mass varies from study to study. The majority of studies defined relative skeletal muscle mass by dividing SMM by body weight; however, some chose to divide it by height or BMI [20]. Unifying the definition of relative skeletal muscle mass in future MASLD studies is crucial.

In this health check-up cohort, the prevalence of sarcopenia by the JSH criteria in MASLD was very low. According to a previous report, the prevalence of sarcopenia was 17.9–43.6% in a biopsy-proven NAFLD cohort [18,19]. In these studies, sarcopenia was defined as an appendicular skeletal muscle mass/weight value that falls more than one or two standard deviations below the average value for healthy young adults. These criteria vary considerably from the liver disease-specific secondary sarcopenia criteria by the JSH. In addition, the prevalence of sarcopenia may differ between the biopsy-proven and health checkup cohorts. The definition of sarcopenia in MASLD should be unified and the use of SMM/WT for this definition would be useful.

MASLD, sarcopenia, and obesity are closely interconnected, and each plays a role in the pathology of the others. Recent studies have highlighted sarcopenia as a novel risk factor for NAFLD [21]. Sarcopenia rates are notably higher in individuals with NAFLD/NASH than in healthy controls. Sarcopenia is linked to significant fibrosis and steatosis, independent of hepatic and metabolic risk factors [18,19]. Wang et al. found that a reduction in both muscle mass and strength is related to an elevated risk of developing NAFLD [22]. A meta-analysis showed that patients with sarcopenia have a significantly higher risk of developing NAFLD, NASH, and NAFLD-related fibrosis [20]. A large cohort study involving 333,295 individuals in the United Kingdom found that lower muscle mass was linked to an increased risk of severe NAFLD [23]. Although the five cardiometabolic criteria for MASLD do not include an assessment of body composition components, we strongly recommend muscle mass assessment in individuals with MASLD, and SMM/WT should be used to assess muscle mass.

This study has several limitations. A cohort from a single health checkup center was used for this study. Differences in age, gender, and ethnic background may affect the body composition components. Therefore, further analysis should include various cohorts in a multicenter setting. Second, the analysis did not include biochemical data (blood glucose, HbA1c, blood pressure, triglycerides, and HDL). The reason for excluding biochemical data is that the medications could normalize them. Biochemical data play such an important role in the diagnosis of disease and the evaluation of predictive factors. Their absence may limit the interpretation of results.

## 5. Conclusions

We found that SMM/WT was significantly associated with MASLD. Therefore, SMM/WT may serve as a potential marker for the diagnosis of MASLD. Unifying the definition of relative skeletal muscle mass and sarcopenia in MASLD is needed in the future; we strongly recommend the use of SMM/WT for these assessments.

## Figures and Tables

**Figure 1 nutrients-16-04422-f001:**
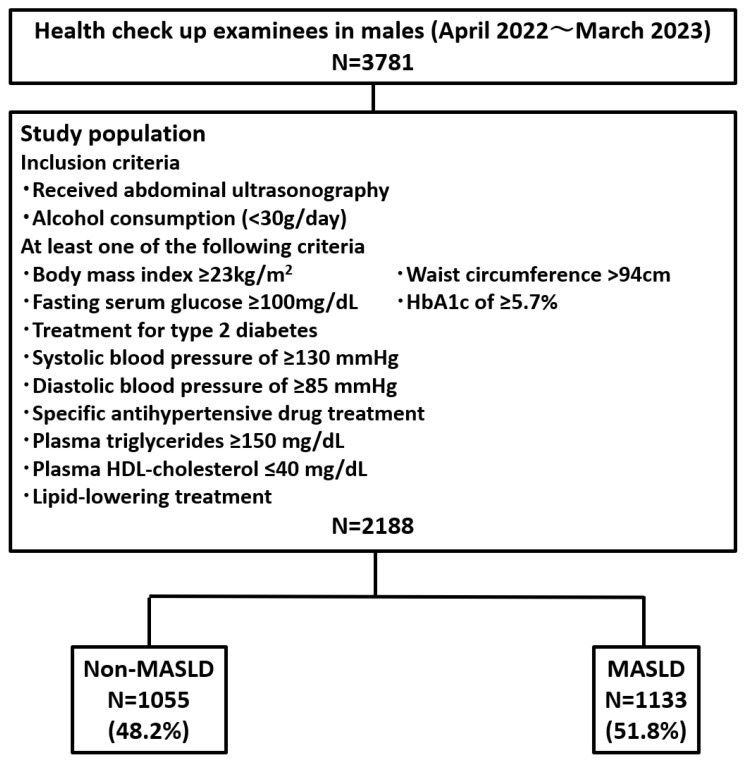
Diagram illustrating the study flow. We considered 6599 examinees. N, number; HbA1C, glycated hemoglobin; HDL, high-density lipoprotein; MASLD, metabolic dysfunction-associated steatotic liver disease.

**Figure 2 nutrients-16-04422-f002:**
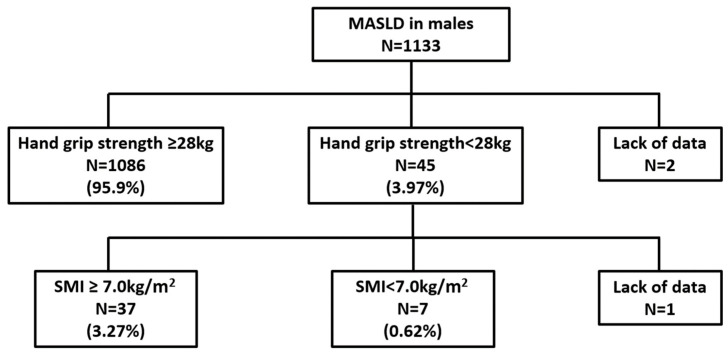
Prevalence of sarcopenia among health checkup examinees. Sarcopenia was identified in 0.62% of males and 0.66% of females with MASLD by the JSH (2nd edition) sarcopenia criteria. SMI, skeletal muscle mass index.

**Table 1 nutrients-16-04422-t001:** Baseline characteristics of the non-MASLD and MASLD groups. The values are presented as the median (interquartile range). The Wilcoxon signed-rank test regarded a *p*-value < 0.005 as significant. MASLD, metabolic dysfunction-associated steatotic liver disease; BMI, body mass index; WC, waist circumference; TBW, total body water; ICW, intracellular water; ECW, extracellular water; BFM, body fat mass; SLM, soft lean mass; FFM, fat-free mass; SMM, skeletal muscle mass; PBF, percentage body fat; SMI, skeletal muscle mass index; VFA, visceral fat area; SMM/WT, skeletal muscle mass/weight; FFMI, fat-free mass index; FMI, fat mass index.

Males				Females			
	Non–MASLD (*n* = 1055)	MASLD (*n* = 1133)	*p*–Value		non–MASLD (*n* = 1222)	MASLD (*n* = 454)	*p*–Value
Age (years)	53.0 (45.0–60.0)	54.0 (47.0–59.0)	n.s	Age (years)	52.0 (46.0–58.0)	54.0 (48.0–59.0)	0.0012
Height (cm)	170.5 (166.4–174.4)	170.4 (166.6–174.3)	n.s	Height (cm)	157.7 (154.1–161.3)	157.4 (153.6–160.8)	n.s
Weight (kg)	65.9 (60.4–71.7)	74.3 (67.9–82.8)	<0.0001	Weight (kg)	53.3 (48.1–59.1)	65.6 (59.0–73.2)	<0.0001
BMI (kg/m^2^)	22.9 (21.1–24.3)	25.7 (23.8–28.1)	<0.0001	BMI (kg/m^2^)	21.5 (19.6–23.5)	26.3 (23.8–29.7)	<0.0001
WC (cm)	81.5 (76.8–86.1)	89.8 (84.6–96.4)	<0.0001	WC (cm)	77.6 (71.5–83.5)	89.0 (83.7–96.5)	<0.0001
TBW (L)	37.8 (35.3–40.8)	40.0 (37.1–43.6)	<0.0001	TBW (L)	27.3 (25.6–29.4)	29.7 (27.5–32.0)	<0.0001
ICW (L)	23.5 (21.8–25.3)	24.9 (23.1–27.1)	<0.0001	ICW (L)	16.8 (15.7–18.1)	18.3 (16.9–19.7)	<0.0001
ECW (L)	14.4 (13.4–15.5)	15.1 (14.0–16.5)	<0.0001	ECW (L)	10.6 (9.9–11.3)	11.4 (10.6–12.3)	<0.0001
Protein (kg)	10.2 (9.4–10.9)	10.7 (10.0–11.7)	<0.0001	Protein (kg)	7.3 (6.8–7.8)	7.9 (7.3–8.5)	<0.0001
Minerals (kg)	3.42 (3.17–3.70)	3.64 (3.37–3.99)	<0.0001	Minerals (kg)	2.56 (2.42–2.78)	2.79 (2.55–3.02)	<0.0001
BFM (kg)	14.0 (11.0–17.4)	20.0 (16.2–25.0)	<0.0001	BFM (kg)	16.2 (12.6–20.0)	24.8 (20.5–30.6)	<0.0001
SLM (kg)	48.6 (45.3–52.5)	51.4 (47.7–56.0)	<0.0001	SLM (kg)	35.0 (32.8–37.7)	38.0 (35.2–41.0)	<0.0001
FFM (kg)	51.4 (47.9–55.5)	54.4 (50.5–59.2)	<0.0001	FFM (kg)	37.2 (34.8–40.0)	40.4 (37.3–43.5)	<0.0001
SMM (kg)	28.6 (26.4–31.0)	30.4 (28.1–33.4)	<0.0001	SMM (kg)	19.9 (18.5–21.6)	21.8 (20.0–23.7)	<0.0001
PBF (%)	21.3 (17.7–25.0)	27.0 (23.4–31.1)	<0.0001	PBF (%)	30.4 (25.4–34.5)	38.0 (34.3–42.1)	<0.0001
SMI (kg/m^2^)	7.6 (7.1–8.0)	7.9 (7.5–8.5)	<0.0001	SMI (kg/m^2^)	5.9 (5.6–6.4)	6.6 (6.2–7.1)	<0.0001
VFA (cm^2^)	59.6 (46.6–75.5)	86.9 (69.1–110.9)	<0.0001	VFA (cm^2^)	71.5 (54.7–98.1)	125.6 (97.4–160.9)	<0.0001
SMM/WT (%)	43.8 (41.7–45.9)	40.8 (38.3–42.9)	<0.0001	SMM/WT (%)	37.3 (34.9–39.8)	33.5 (31.3–35.4)	<0.0001
FFMI (kg/m^2^)	17.7 (16.8–18.8)	18.8 (17.8–19.9)	<0.0001	FFMI (kg/m^2^)	15.0 (14.2–15.8)	16.3 (15.4–17.5)	<0.0001
FMI (kg/m^2^)	4.8 (3.8–6.0)	6.9 (5.6–8.7)	<0.0001	FMI (kg/m^2^)	6.5 (5.1–8.0)	9.9 (8.3–12.6)	<0.0001
Hand grip strength (kg)	36.8 (33.0–40.1)	37.2 (33.6–41.4)	0.0019	Hand grip strength (kg)	23.6 (21.2–25.9)	24.1 (21.9–26.9)	0.0007

Values are expressed as the median (interquartile range). n.s.: no significance.

**Table 2 nutrients-16-04422-t002:** Multivariate analysis of the risk factors associated with MASLD.

	Males			Females		
	Multivariate Analysis			Multivariate Analysis		
	OR	95%CI	*p*–Value	OR	95%CI	*p*–Value
BMI	1.05	0.89–1.24	0.5367	1.00	0.84–1.20	0.9875
WC	1.09	1.05–1.13	<0.0001	1.05	1.02–1.08	0.0004
TBW	1.54	0.99–2.38	0.0529	0.70	0.38–1.26	0.2307
Protein	0.15	0.03–0.78	0.0242	1.97	0.19–20.35	0.5683
PBF	2.78	2.03–3.79	<0.0001	2.24	1.51–3.33	<0.0001
SMI	0.77	0.40–1.48	0.4382	2.34	0.97–5.65	0.0589
SMM/WT	5.11	2.99–8.73	<0.0001	3.41	1.68–6.93	0.0006
Hand grip strength	1.01	0.99–1.03	0.3458	1.00	0.96–1.05	0.8293

OR, odds ratio; 95%CI, 95% confidence interval.

## Data Availability

The data supporting the findings of this study are available from the corresponding author upon reasonable request.

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
