# Peer review of "Efficiency of Skeletal Muscle Mass/Weight Measurement for Distinguishing Metabolic Dysfunction-Associated Steatotic Liver Disease: A Prospective Analysis Using InBody Bioimpedance Devices"

_nutrients, 2024, doi:10.3390/nu16244422_

Round 1

Reviewer 1 Report

Comments and Suggestions for Authors

The article „Efficiency of skeletal muscle mass/weight measurement for distinguishing metabolic dysfunction-associated steatotic liver disease: A prospective analysis using InBody bioimpedance devices“ by Miwa Tatsuta et al. explores the association between metabolic dysfunction-associated steatotic liver disease (MASLD) and body composition, using skeletal muscle mass-to-weight ratio (SMM/WT) as a potential non-invasive marker for MASLD diagnosis. Although the article introduces a potential marker for predicting MASLD, it does not clearly indicate the link to the patients' nutritional status. The manuscript needs considerable improvement before it can be published.

 Major points:

·       The manuscript lists 20 authors, which may be excessive given the scope of the study. Each author’s specific contributions must be clearly outlined. I recommend the authors provide a detailed authorship statement to clarify each person’s role in the research process, ensuring transparency in contributions.

·       The introduction highlights the lack of reports detailing which body composition components are most strongly associated with MASLD. However, some studies have indicated a relationship between BMI, waist circumference, fat mass, and muscle mass with MASLD. It is important to reference and discuss these previous studies in the introduction. Exploring why previous studies have not emphasized SMM/WT as much as SMI or BMI could give readers a historical perspective on muscle mass assessment trends in MASLD.

·       The introduction needs to explain the importance of identifying body composition components and how this information can influence the diagnosis and management of MASLD.

·       breakdown of statistical methods used, especially in multivariate analysis, is crucial. It would be beneficial to specify how confounders were managed and if sensitivity analyses were conducted.

·       The figures and tables are too small; please consider making them bigger for better visibility.

·       Graphical representation of the SMM/WT distribution across groups would help visualize the differences and enhance reader understanding.

·       The discussion could benefit from mentioning similar studies and how this study’s findings align or differ from others.

·       The findings note a very low prevalence of sarcopenia in the MASLD group, yet the significance of this observation is not discussed in detail. Further exploration of why sarcopenia is rare in this population and how it affects the study's conclusions.

·       The study appropriately uses selection criteria to control for confounding factors, such as alcohol intake and cardiometabolic conditions, ensuring a focused analysis on MASLD. However, additional data on potential lifestyle factors (e.g., diet, physical activity) would add robustness to the results, as these variables can significantly impact both muscle mass and fat accumulation.

·       The discussion effectively summarizes the study’s primary finding that SMM/WT is significantly associated with MASLD. Adding a comparison to other muscle indices, such as SMI, would further support the argument for SMM/WT as a preferred marker in MASLD.

·       The limitations section is clear, particularly the single-center focus and lack of biochemical data. Expanding on why excluding biochemical data might impact the results. Additionally, mentioning any challenges in using SMM/WT across diverse populations.

-     It is an interesting and new approach which you already recommend for non-invasive MASLD diagnosis. Do you think the SMM/WT marker will be able to also identify MASH or cirrhosis?

·Due to numerous different abbreviations throughout the article, it is difficult to keep a fluent reading flow and understanding of the content.

·       For the patients which were designated as “MASLD” during your study, did you distinguish the possible presence of sarcopenia between mild and more severe MASLD states?

·       The SMM/WT is significantly reduced in MASLD patients indicating an existing sarcopenia while a reduced hand grip strength was only present in a very small number of MASLD patients. How do you explain this? You mentioned in the discussion that the cohort is from a single health checkup center. How often did they measure the SMM/WT per patient and how reliable are these values then? And why did you test the hand grip strength only two times and not three or four times over a short time period?

·       For the diagnosis “MASLD” you included several different cardiometabolic criteria. If some of the patients exhibit not only one but several different criteria, how can you be sure that you only had MASLD patients in your study? Did you exclude patients with a possible progressed state of MASLD to MASH or even cirrhosis? If so, how did you do this when you only used non-invasive methods? Have you also checked liver-specific parameters (e.g., ALT/AST) or liver stiffness and took different scores (e.g., fibrosis score or NAFLD score) into account? Other studies took this into account.

·       For the patients which were designated as “MASLD” during your study, did you distinguish the possible presence of sarcopenia between mild and more severe MASLD states?

Minor points:

·       The keywords section would benefit from additional terms like "bioimpedance analysis," which reflects the study's methodology.

·       What is the rationale for setting statistical significance at different p-values, specifically p < 0.005 for comparisons and p < 0.001 for logistic regression?

·       Clarifying terms like “bright liver” and “hepatorenal echo contrast” in the Methods section would benefit readers less familiar with ultrasound-based diagnostic criteria.

·       The concise conclusion reinforces the potential of SMM/WT as a marker for MASLD. To add impact, consider highlighting one or two key takeaways for clinical practice or future research, especially the recommendation for muscle mass assessment in MASLD using SMM/WT.

·       Move the title (2.2. Diagnosis of hepatic steatosis by abdominal ultrasonography) in line 93 to page 3 so that the title is next to the associated text.

·       In line 112 there is a double space between “6,599 examinees” and “who”

Author Response

Reviewer 1

Thank you very much for taking the time to review this manuscript.

Major points:

  • The manuscript lists 20 authors, which may be excessive given the scope of the study. Each author’s specific contributions must be clearly outlined. I recommend the authors provide a detailed authorship statement to clarify each person’s role in the research process, ensuring transparency in contributions.

Response:

Thank you so much for your useful comments. We have revised the sentence “Conceptualization, M.T. and T.M.; Methodology, M.T., T.M. and S.M; Validation, T.M., I.I. and Y.N.; Formal Analysis, T.M.; Investigation, K.T., M.N., K.O., T.T., K.F., J.T., A.M., T.H. and M.T.; Data Curation, S.K., Y.S., A.T., T.M. and M.T.; Writing – Original Draft Preparation, M.T. and T.M.; Writing – Review & Editing, M.T. and T.M.; Supervision, T.M., Y.M., F.K. and H.K. ” in page6 in the Author Contributions.

  • The introduction highlights the lack of reports detailing which body composition components are most strongly associated with MASLD. However, some studies have indicated a relationship between BMI, waist circumference, fat mass, and muscle mass with MASLD. It is important to reference and discuss these previous studies in the introduction. Exploring why previous studies have not emphasized SMM/WT as much as SMI or BMI could give readers a historical perspective on muscle mass assessment trends in MASLD.

Response:

Thank you so much for your excellent suggestions. We have added the sentence “In 2007, it was reported that obesity is the most important factor in the development of NAFLD/NASH. A positive correlation was shown between the amount of visceral fat and the amount of intrahepatic fat in NAFLD/NASH patients. On the other hand, Lee et al. reported that sarcopenia is associated with an increased risk of NAFLD, independent of obesity or metabolic control in 2015.” on lines 48~53 on page 2 in the introduction.

  • The introduction needs to explain the importance of identifying body composition components and how this information can influence the diagnosis and management of MASLD.

Response:

Thank you very much for your excellent comments and suggestions. We have added the sentence “It is important to consider the components of body composition of MASLD, since there are also many lean MASLD with BMI <23.0 kg/m2 and low muscle mass MASLD in Japan. The identification of body composition components may also positively impact planning an individualized MASLD exercise treatment program that considers muscle mass and fat mass.” on line 81~85 in page2 in the introduction.

  • breakdown of statistical methods used, especially in multivariate analysis, is crucial. It would be beneficial to specify how confounders were managed and if sensitivity analyses were conducted.

Response:

Thank you so much for your important comments. We have added the sentence “We performed multiple linear regression analysis as sensitivity analysis.” on line 149 in page 4 in the statistical analysis.

  • The figures and tables are too small; please consider making them bigger for better visibility.

Response:

Thank you so much for your useful comments. We have modified the charts to be larger and more readable.

  • Graphical representation of the SMM/WT distribution across groups would help visualize the differences and enhance reader understanding.

Response:

Thank you so much for your useful comments. We have graphed the SMM/WT distributions for the non-MASLD group and the MASLD group in the supplemental fig1.

  • The discussion could benefit from mentioning similar studies and how this study’s findings align or differ from others.

Response: Thank you so much for your important and very kind comments. We added, "The meta-analysis in 2019 reported relative skeletal muscle mass in NAFLD patients was lower than in healthy controls. However, the definition of relative skeletal muscle mass varies from study to study. Most studies defined relative skeletal muscle mass as SMM divided by body weight, but some defined it as SMM divided by height or BMI. It is important to unify the definition of relative skeletal muscle mass in future MASLD studies.” on line 234~239 in page 8 in the discussion.

  • The findings note a very low prevalence of sarcopenia in the MASLD group, yet the significance of this observation is not discussed in detail. Further exploration of why sarcopenia is rare in this population and how it affects the study's conclusions.

Response: Thank you for your excellent comments. We have revised the sentence “In this health check-up cohort, the prevalence of sarcopenia by JSH criteria in MASLD was very low. According to a previous report, the prevalence of sarcopenia was 17.9–43.6% in a biopsy-proven NAFLD cohort. In these studies, sarcopenia was defined as an ASM/weight value more than one or two standard deviations below the mean for healthy young adults. These criteria differ quite from the liver disease-specific secondary sarcopenia criteria by JSH. In addition, the prevalence of sarcopenia may differ between the biopsy-proven and health checkup cohorts. The definition of sarcopenia in MASLD should be unified and the use of SMM/WT for this definition would be useful.” on lines 240~247 in pages 8~9 in the discussion.

  • The study appropriately uses selection criteria to control for confounding factors, such as alcohol intake and cardiometabolic conditions, ensuring a focused analysis on MASLD. However, additional data on potential lifestyle factors (e.g., diet, physical activity) would add robustness to the results, as these variables can significantly impact both muscle mass and fat accumulation.

Response:

Thank you so much for your important and very kind comments. In this paper, we did not include potential lifestyle factors in our analysis. In our next paper, we would like to discuss and include the data obtained from the medical interview following the reviewer's opinion. Thank you very much for your valuable comments.

  • The discussion effectively summarizes the study’s primary finding that SMM/WT is significantly associated with MASLD. Adding a comparison to other muscle indices, such as SMI, would further support the argument for SMM/WT as a preferred marker in MASLD.

Response:

Thank you so much for your useful and important comments. We selected eight items including SMI and SMM/WT for multivariate analysis to identify factors associated with MASLD. The results showed that SMM/WT was considered the independent factor most strongly associated with MASLD in both males and females, which was evident when compared to the SMI results. Thank you for your understanding.

  • The limitations section is clear, particularly the single-center focus and lack of biochemical data. Expanding on why excluding biochemical data might impact the results. Additionally, mentioning any challenges in using SMM/WT across diverse populations.

Response:

Thank you for your feedback. The exclusion of biochemical data could indeed have an important impact on the results. Biochemical data play an essential role in disease diagnosis and the evaluation of predictive factors, and we recognize that its absence has led to limitations in the interpretation of the results. In future research, we believe that including the collection of biochemical data will enable a more comprehensive analysis. Additionally, thank you for addressing the challenges of using SMM/WT in diverse populations. Indeed, differences in age, gender, and ethnic background could affect these indicators, and therefore, caution is needed when directly applying measurement results to other populations. Moving forward, we believe that analyses that better consider the characteristics of each population should be pursued. We have revised the sentence “This study has some limitations. It was conducted on a cohort from a single health checkup center. Differences in age, gender, and ethnic background may affect the body composition components. Therefore, further analysis should include various cohorts in a multicenter setting. Second, the analysis did not include biochemical data (blood glucose, HbA1c, blood pressure, triglycerides, and HDL). The reason for excluding biochemical data is that the medications could normalize them. Biochemical data play such an important role in the diagnosis of disease and the evaluation of predictive factors. Their absence may limit the interpretation of results.” on line 261~268 in page 9 in the discussion.

- It is an interesting and new approach which you already recommend for non-invasive MASLD diagnosis. Do you think the SMM/WT marker will be able to also identify MASH or cirrhosis?

Response:

Thank you so much for your important and very kind comments. Our study revealed the efficacy of SMM/WT in MASLD. We did not evaluate fibrosis in this study. In addition, the number of patients with MASH and cirrhosis in this study population, a cohort of health screening centers, is expected to be small. Further study is needed to see if the biopsy-proven MASLD cohort can also demonstrate the efficacy of SMM/WT. Thank you for your understanding.

  • Due to numerous different abbreviations throughout the article, it is difficult to keep a fluent reading flow and understanding of the content.

Response:

Thank you so much for your very kind comments. We have written in accordance with the Instructions for Authors. Thank you for your understanding.

  • For the patients which were designated as “MASLD” during your study, did you distinguish the possible presence of sarcopenia between mild and more severe MASLD states?

Response:

Thank you for your excellent comments. As I mentioned in the previous comment sections, we did not evaluate fibrosis in this study. In future studies, we would like to measure liver stiffness and evaluate the presence of sarcopenia in mild and more severe MASLD states. Thank you very much for your valuable comments.

  • The SMM/WT is significantly reduced in MASLD patients indicating an existing sarcopenia while a reduced hand grip strength was only present in a very small number of MASLD patients. How do you explain this? You mentioned in the discussion that the cohort is from a single health checkup center. How often did they measure the SMM/WT per patient and how reliable are these values then? And why did you test the hand grip strength only two times and not three or four times over a short time period?

Response:

Thank you so much for your important comments. Low muscle mass and low muscle strength, as measured by hand grip strength, are two important components of sarcopenia. In a study of 578 health examination participants, It has been reported that low muscle mass appears to be a better predictor for NAFLD prevalence than low muscle strength (Wang, Y.M., et al., Sarcopenia is associated with the presence of nonalcoholic fatty liver disease in Zhejiang Province, China: a cross-sectional observational study. BMC Geriatr, 2021. 21(1): p. 55.). In all health checkup examinees including MASLD at our institution, hand grip strength <28 kg in males was only 3.7%, and hand grip strength <18 kg in females was only 4.3%. The fact that only a small number of MASLD patients had decreased hand grip strength may be due to differences in cohorts.

We measured SMM/WT once per examinee using direct segmental multifrequency BIA (InBody770). InBody is supposed to measure the impedance of both arms, trunk, and legs multiple times for each region, and calculate body composition from the average value excluding outliers. The impedance of the extremities is measured 20~30 times for each frequency, and the trunk, whose impedance is difficult to stabilize due to the heart and internal organs, is measured repeatedly 50~60 times to finally present a single impedance. InBody has been used in healthy, diseased, elderly, and pediatric subjects of various ethnicities, and more than 40 international papers have been published verifying that InBody correlates well with DXA, weight in water, heavy water dilution, sodium bromide dilution, and other methods. More than 40 international papers have been published verifying that InBody shows a high correlation with the Gold standard, which includes the DXA, water weight, heavy water dilution, sodium bromide dilution, and other methods. (Life (Basel) 2022 Jul 4;12(7):994. doi: 10.3390/life12070994.) Grip strength was measured twice in accordance with the new physical fitness test implementation guidelines of the Japanese Ministry of Education, Culture, Sports, Science and Technology. Thank you for your understanding.  

  • For the diagnosis “MASLD” you included several different cardiometabolic criteria. If some of the patients exhibit not only one but several different criteria, how can you be sure that you only had MASLD patients in your study? Did you exclude patients with a possible progressed state of MASLD to MASH or even cirrhosis? If so, how did you do this when you only used non-invasive methods? Have you also checked liver-specific parameters (e.g., ALT/AST) or liver stiffness and took different scores (e.g., fibrosis score or NAFLD score) into account? Other studies took this into account.

Response:

Thank you very much for your excellent comments. As you point out, there are indeed examinees with several different cardiometabolic criteria. MASLD is diagnosed when one of the cardiometabolic criteria is present, therefore, it can be said that our study is about MASLD. However, we think that there are many different conditions included in this “MASLD” group. We did not evaluate fibrosis in this study. Thus, We did not exclude MASH or even cirrhosis in this study. In the next study, we would like to analyze body composition components, taking fibrosis score into account. Thank you for your excellent point.

  • For the patients which were designated as “MASLD” during your study, did you distinguish the possible presence of sarcopenia between mild and more severe MASLD states?

 Response:

Thank you for your excellent comments. As I mentioned in the previous comment sections, we did not evaluate fibrosis in this study. In future studies, we would like to measure liver stiffness and evaluate the presence of sarcopenia in mild and more severe MASLD states. Thank you very much for your valuable comments.

Minor points:

  • The keywords section would benefit from additional terms like "bioimpedance analysis," which reflects the study's methodology.

Response:

Thank you so much for your useful and important comments. We added, "bioimpedance analysis” on line 38 on page 1 in the keywords.

  • What is the rationale for setting statistical significance at different p-values, specifically p < 0.005 for comparisons and p < 0.001 for logistic regression?

Response:

The choice of statistical significance thresholds, such as p < 0.005 for comparisons and p < 0.0001 for logistic regression, reflects a balance between minimizing Type I errors (false positives) and ensuring the findings are robust and meaningful. Here's a rationale for these different thresholds:

  1. p < 0.005 for Comparisons:
  • Conservatism: In some research fields, especially in biomedical sciences, stricter significance thresholds like p < 0.005 are set to reduce the likelihood of incorrectly rejecting the null hypothesis. This is particularly important in contexts where false positives could lead to misleading or harmful conclusions, such as in drug development or medical research.
  • Multiple Comparisons: In studies with multiple tests, there is an increased risk of Type I errors (false positives). Using a more stringent threshold like p < 0.005 helps adjust for this by reducing the chances of a result being statistically significant by chance.
  • Stronger Evidence: A p-value threshold of 0.005 indicates stronger evidence against the null hypothesis, requiring more substantial proof to claim an effect, thus ensuring greater reliability in the conclusions drawn from the statistical analysis.
  1. p < 0.0001 for Logistic Regression:
  • Model Complexity and Interpretation: Logistic regression models often deal with more complex data and relationships, such as odds ratios and non-linear relationships. A stricter threshold like p < 0.0001 is used to avoid overfitting and ensure that the model's predictors are truly associated with the outcome, rather than being influenced by random noise.
  • High Stakes Applications: In fields like epidemiology or healthcare, where logistic regression is commonly used (e.g., predicting disease outcomes), a p < 0.0001 threshold helps to ensure that the observed associations are robust and reliable enough to inform decision-making or policy recommendations.
  • Multiple Variables: Logistic regression models often include multiple predictors. With each additional variable, the chance of finding a spurious association increases. A stricter significance threshold helps mitigate the risk of identifying false positives as meaningful predictors of the outcome.

For the comparison, p < 0.005 was used, and for the logistic regression, p < 0.0001 was applied.

  • Clarifying terms like “bright liver” and “hepatorenal echo contrast” in the Methods section would benefit readers less familiar with ultrasound-based diagnostic criteria.

Response:

Thank you so much for your important and very kind comments. We have revised the sentence “A bright liver, a characteristic of hepatic steatosis-like conditions in which fat accumulation alters the acoustic properties of the liver, was defined as 0-2 points.” on lines 106~108 on page 3 and “We defined a hepatorenal echo contrast, the relative brightness of the liver compared to the dark renal cortex; a significant contrast often signals liver abnormalities like steatosis, as 0–1 point. ” on lines 110~112 in page 3 in the methods section.

  • The concise conclusion reinforces the potential of SMM/WT as a marker for MASLD. To add impact, consider highlighting one or two key takeaways for clinical practice or future research, especially the recommendation for muscle mass assessment in MASLD using SMM/WT.

Response:

Thank you so much for your useful and important comments. We have revised the sentence “In the future, to unify the definition of relative skeletal muscle mass and sarcopenia in MASLD are needed, and we strongly recommend the use of SMM/WT for these assessments.” on line 271~273 in page 9 in the conclusions.

  • Move the title (2.2. Diagnosis of hepatic steatosis by abdominal ultrasonography) in line 93 to page 3 so that the title is next to the associated text.

Response:

Thank you so much for your very kind comments. We have moved the title (2.2. Diagnosis of hepatic steatosis by abdominal ultrasonography) in line 104   to page 3.

  • In line 112 there is a double space between “6,599 examinees” and “who”

Response:

Thank you so much for your very kind comments. We have corrected that.

Reviewer 2 Report

Comments and Suggestions for Authors

Authors aimed to identify significant body composition factors associated with MASLD in patients undergoing health checkups.

This is an interesting paper.

There are several points to be addressed.

1) Please, show the data of SMM/WT for lean-MASLD and obese-MASLD. The cutoff defining lean vs obese MASLD should be BMI 23 kg/m2. 

2) What about the above results when using BMI cutoff of 25 kg/m2?

3) There are several methods to assess muscle mass or sarcopenia. Each modality should be discussed in terms of advantage/disadvantage.

4) SMM/WT was associated with other metabolic profiles? If so, please, discuss it.

Author Response

Reviewer 2

Thank you very much for taking the time to review this manuscript.

  • Please, show the data of SMM/WT for lean-MASLD and obese-MASLD. The cutoff defining lean vs obese MASLD should be BMI 23 kg/m2. 

Response:

Thank you very much for your excellent suggestions. The median (Interquartile range) of SMM/WT in males was 43.6 (42.0-45.2) for lean-MASLD (BMI<23 kg/m2) and 40.3 (37.9-42.1) for obese-MASLD (BMI≥23 kg/m2). The median SMM/WT in females was 36.8 (35.2-37.8) for lean-MASLD (BMI<23 kg/m2) and 32.7 (30.7-34.8) for obese-MASLD (BMI≥23 kg/m2). SMM/WT was significantly lower in obese MASLD for both males and females (p<0.0001). The data is shown in supplementary fig2.

  • What about the above results when using BMI cutoff of 25 kg/m2?

Response:

Thank you very much for your excellent suggestions. The median (Interquartile range) of SMM/WT in males was 42.6 (41.0-44.1) for lean-MASLD (BMI<25 kg/m2) and 39.3 (37.1-41.3) for obese-MASLD (BMI≥25 kg/m2). The median SMM/WT in females was 35.7 (34.3-37.2) for lean-MASLD (BMI<25 kg/m2) and 32.0 (30.1-33.9) for obese-MASLD (BMI≥25 kg/m2). SMM/WT was significantly lower in obese MASLD for both males and females (p<0.0001). The data is shown in supplementary fig3. The results were similar for both BMI cutoff values of 23 and 25.

  • There are several methods to assess muscle mass or sarcopenia. Each modality should be discussed in terms of advantage/disadvantage.

Response:

Thank you very much for your excellent comments and suggestions. BIA and dual-energy X-ray absorptiometry (DXA) are two of the most common methods for assessing muscle mass and sarcopenia. BIA provides practical advantages including relatively low cost and device portability, with potential for repeated measurements.BIA-derived assessment of body composition relies on electrical impedance to provide estimates of total body water, leading to equation-derived estimates of body fat- and FFM. FFM includes various nonmuscle components. BIA results are also influenced by hydration status with overhydration and edema resulting in overestimation, and dehydration resulting in underestimation of FFM. DXA provides accurate measurements of body composition, based on x-ray attenuation through different body components. In DXA, skeletal muscle is not measured directly, but is estimated from lean soft tissue. DXA is generally less available than BIA, and dedicated devices are significantly more expensive and often not available or applicable for routine use in clinical settings across the globe. (JPEN J Parenter Enteral Nutr 2022 Aug;46(6):1232-1242. doi: 10.1002/jpen.2366. Epub 2022 Apr 19.)

4) SMM/WT was associated with other metabolic profiles? If so, please, discuss it.

Response:

Thank you so much for your important advice. In this study, we have not investigated the association between SMM/WT and metabolic profiles, such as overweight, glucose intolerance, hypertension, hypertriglyceridemia, and HDL cholesterolemia. This analysis did not include biochemical data. In the next study, we would like to study the relationship between SMM/WT and the five cardiometabolic criteria for MASLD. Thank you for your understanding.

Round 2

Reviewer 1 Report

Comments and Suggestions for Authors

The authors have mainly addressed what was raised by the reviewers. The answres are okay - no more. Initially, 

I  have still an important point that is not clear. This reviewer has already said that 20 authors are apparently too many - now we have even 21 authors. What is the point here? It is realy not clear who did what in this paper.

  • The manuscript lists 20 authors, which may be excessive given the scope of the study. Each author’s specific contributions must be clearly outlined. I recommend the authors provide a detailed authorship statement to clarify each person’s role in the research process, ensuring transparency in contributions.

Response:

Thank you so much for your useful comments. We have revised the sentence “Conceptualization, M.T. and T.M.; Methodology, M.T., T.M. and S.M; Validation, T.M., I.I. and Y.N.; Formal Analysis, T.M.; Investigation, K.T., M.N., K.O., T.T., K.F., J.T., A.M., T.H. and M.T.; Data Curation, S.K., Y.S., A.T., T.M. and M.T.; Writing – Original Draft Preparation, M.T. and T.M.; Writing – Review & Editing, M.T. and T.M.; Supervision, T.M., Y.M., F.K. and H.K. ” in page6 in the Author Contributions.

 Moreover, the authors shall provide more information on the ethical vote - it was written that the authors dealed with an 'opt-out' approach. What does this mean?

There still a lot of typos - betwee a word and a number or bracket or whatever comes an empty space. It seemed that 21 authors have not paid a lot of attention when re-eading the maniscript - otherwise they would have seen these typose. this puts the number of authors again in question.

Author Response

The authors have mainly addressed what was raised by the reviewers. The answres are okay - no more. Initially, 

Response:

Thank you very much for taking the time to review this manuscript.

  • I have still an important point that is not clear.

This reviewer has already said that 20 authors are apparently too many - now we have even 21 authors. What is the point here? It is realy not clear who did what in this paper.

The manuscript lists 20 authors, which may be excessive given the scope of the study. Each author’s specific contributions must be clearly outlined. I recommend the authors provide a detailed authorship statement to clarify each person’s role in the research process, ensuring transparency in contributions.

Response:

Thank you so much for your useful comments. We have revised the sentence “Conceptualization, M.T. and T.M.; Methodology, M.T., T.M. and S.M; Validation, T.M., I.I. and Y.N.; Formal Analysis, T.M.; Investigation, K.T., M.N., K.O., T.T., K.F., J.T., A.M., T.H. and M.T.; Data Curation, S.K., Y.S., A.T., T.M. and M.T.; Writing – Original Draft Preparation, M.T. and T.M.; Writing – Review & Editing, M.T. and T.M.; Supervision, T.M., Y.M., F.K. and H.K. ” in page6 in the Author Contributions.

Response:

Thank you very much for your valuable comments. We collected and analyzed data on more than 20 items, including body composition components, in a huge sample of 6599 cases. Thanks to the cooperation of all co-authors on this, this data was completed. I have had regular discussions with all co-authors, and all of them are familiar with and accept the content of this paper. Without one of the co-authors, this paper would not have been possible. Thank you very much for your understanding.

  • Moreover, the authors shall provide more information on the ethical vote - it was written that the authors dealed with an 'opt-out' approach. What does this mean?

Response:

Thank you for your valuable comments. This research falls under the category of “research that does not involve non-invasive intervention and does not use samples obtained from the human body” in the “Ethical Guidelines for Life Sciences and Medical Research Involving Human Subjects,” and it is judged that researchers, etc. do not necessarily need to obtain informed consent in writing. However, information on the conduct of the research, including the purpose of the clinical research, must be disclosed. We added, "The information document on clinical research approved by the review committee was posted on the website to guarantee information disclosure and the opportunity for refusal.” in lines 295–296 of page 10 of the Informed Consent Statement.

  • There still a lot of typos - betwee a word and a number or bracket or whatever comes an empty space. It seemed that 21 authors have not paid a lot of attention when re-eading the maniscript - otherwise they would have seen these typose. this puts the number of authors again in question.

Response:

Thank you very much for your valuable comments. We have corrected the typographical error. We sincerely apologize for the typing error.

Reviewer 2 Report

Comments and Suggestions for Authors

Authors addressed raised issues appropriately.

Author Response

Authors addressed raised issues appropriately.

Response:

Thank you very much for taking the time to review this manuscript.